# Interactions between COVID-19 and Lung Cancer: Lessons Learned during the Pandemic

**DOI:** 10.3390/cancers14153598

**Published:** 2022-07-23

**Authors:** David J. H. Bian, Siham Sabri, Bassam S. Abdulkarim

**Affiliations:** 1Faculty of Medicine and Health Sciences, McGill University, Montreal, QC H3G 2M1, Canada; david.bian@mail.mcgill.ca; 2Cancer Research Program, Research Institute, McGill University Health Center Glen Site, McGill University, Montreal, QC H4A 3J1, Canada; siham.sabri.ab@outlook.com; 3Cancer Research Program, Research Institute, and Department of Oncology, Cedars Cancer Center, McGill University Health Center Glen Site, McGill University, Montreal, QC H4A 3J1, Canada

**Keywords:** SARS-CoV-2, COVID-19, COVID-19 therapy, COVID-19 risk factors, lung cancer, cancer therapy, cancer care, inflammation

## Abstract

**Simple Summary:**

COVID-19 is a respiratory infectious disease caused by the coronavirus SARS-CoV-2. Lung cancer is the leading cause of all cancer-related deaths worldwide. As both SARS-CoV-2 and lung cancer affect the lungs, the aim of this narrative review is to provide a consolidation of lessons learned throughout the pandemic regarding lung cancer and COVID-19. Risk factors found in lung cancer patients, such as advanced cancers, smoking, male, etc., have been associated with severe COVID-19. The cancer treatments hormonal therapy, immunotherapy, and targeted therapy have shown no association with severe COVID-19 disease, but chemotherapy and radiation therapy have shown conflicting results. Logistical changes and modifications in treatment plans were instituted during the pandemic to minimize SARS-CoV-2 exposure while maintaining life-saving cancer care. Finally, medications have been developed to treat early COVID-19, which can be highly beneficial in vulnerable cancer patients, with paxlovid being the most efficacious drug currently available.

**Abstract:**

Cancer patients, specifically lung cancer patients, show heightened vulnerability to severe COVID-19 outcomes. The immunological and inflammatory pathophysiological similarities between lung cancer and COVID-19-related ARDS might explain the predisposition of cancer patients to severe COVID-19, while multiple risk factors in lung cancer patients have been associated with worse COVID-19 outcomes, including smoking status, older age, etc. Recent cancer treatments have also been urgently evaluated during the pandemic as potential risk factors for severe COVID-19, with conflicting findings regarding systemic chemotherapy and radiation therapy, while other therapies were not associated with altered outcomes. Given this vulnerability of lung cancer patients for severe COVID-19, the delivery of cancer care was significantly modified during the pandemic to both proceed with cancer care and minimize SARS-CoV-2 infection risk. However, COVID-19-related delays and patients’ aversion to clinical settings have led to increased diagnosis of more advanced tumors, with an expected increase in cancer mortality. Waning immunity and vaccine breakthroughs related to novel variants of concern threaten to further impede the delivery of cancer services. Cancer patients have a high risk of severe COVID-19, despite being fully vaccinated. Numerous treatments for early COVID-19 have been developed to prevent disease progression and are crucial for infected cancer patients to minimize severe COVID-19 outcomes and resume cancer care. In this literature review, we will explore the lessons learned during the COVID-19 pandemic to specifically mitigate COVID-19 treatment decisions and the clinical management of lung cancer patients.

## 1. Introduction

As the Coronavirus-disease-2019 (COVID-19) pandemic enters its third year, the ongoing health crisis has affected every aspect of the healthcare system worldwide. First reported in December 2019 in China, SARS-CoV-2, which stands for “severe acute respiratory syndrome coronavirus 2”, quickly spread to other countries, leading the World Health Organization (WHO) to declare COVID-19 a worldwide pandemic in March 2020. Since then, SARS-CoV-2 infection rates have increased exponentially, on a worldwide scale, with most countries experiencing severe outbreaks in multiple waves. As of 15 July 2022, there have been over 556,000,000 confirmed cases of COVID-19 worldwide, and almost 6,350,000 deaths reported to the WHO, making the COVID-19 pandemic one of the deadliest pandemics in history [1]. The multiple waves of the pandemic were successfully modelled by Kaxiras et al., into a series of epidemic waves (subepidemics) throughout various countries [2]. Certain countries, such as Italy and Portugal, which implemented early strict preventative measures, have shown decreasing peak intensities in subsequent waves. The declining epidemic waves of Italy and Portugal are contrasted with other countries, such as Sweden, which did not impose precautionary measures and have shown epidemic waves increasing in peak intensities in each subsequent subepidemic, highlighting the importance of the early imposition of strict preventative measures to mitigate infection rates in following waves [2]. The need for rapid, stringent public health measures is further stressed by novel variants, such as the Omicron variant, which are shown to be much more transmissible than the wild-type strain, a factor not foreseen by Kaxiras et al. A Chinese modelling study by Cai et al. [3] on Omicron subepidemics, despite China having some of the strictest preventative measures worldwide in pre-Omicron waves, modelled the potentially harsh impact of Omicron on the Chinese healthcare system [3]. Various mitigation strategies, categorized into different scenarios, show that public health measures, such as school and workplace closures, even in the presence of a booster vaccine campaign, were not able to mitigate the impacts of Omicron on the Chinese healthcare system; the peak demand for ICU beds is projected to exceed the maximum capacity in all scenarios simulated. Cai et al. [3] further concluded that there is a need to combine public health measures with increasing vaccine coverage in the elderly and the widespread use of approved antivirals to prevent overwhelming the healthcare system.

Given the importance of public health measures to slow the spread of disease, initial widespread preventative measures, such as lockdown, social distancing, quarantining, and use of personal protective equipment (PPE) mask wearing, were implemented as affected countries attempted to contain SARS-CoV-2 infection rates. In addition to non-pharmaceutical measures of COVID-19 prevention, COVID-19 vaccines, such as the mRNA-based BNT162b2 and mRNA-1273 vaccines, were introduced in December 2020 as a highly effective method of prevention and have been widely approved, distributed, and administered as primary prophylaxis against SARS-CoV-2. As of 15 July 2022, over 12,130,000,000 billion vaccine doses have been administered worldwide [1]. Despite these measures, SARS-CoV-2 outbreaks have continued to occur due to the emergence of novel variants of concern (VOCs), with specific sets of mutations able to increase their infectivity and escape immune response [4]. Additionally, SARS-CoV-2 immunity from vaccination and natural infection is expected to wane over time, allowing milder infections to occur but still providing adequate protection against severe disease [5,6]. We will further discuss these issues in this review. In sum, COVID-19 is expected to be present for the foreseeable future, transitioning into an endemic phase [7].

Given the continuous presence of SARS-CoV-2, it is important to discuss patient populations vulnerable to severe COVID-19 (sCOVID-19), defined as cytokine release syndrome (CRS) and acute respiratory distress syndrome (ARDS) [8]. There is evidence to suggest the risk of COVID-19 being particularly high in cancer patients, especially in lung cancer (LC) patients. Patients with LC are associated with higher rates of mortality when infected with COVID-19 compared to other solid cancers [9,10,11]. This review aims to provide an overview of pathophysiological associations between LC and COVID-19. Additionally, the main risk factors to consider that increase the likelihood of sCOVID-19 in LC patients, as well as the impact of COVID-19 on LC screening, diagnosis, and treatment will be discussed. To minimize sCOVID-19 outcomes in LC patients, it is equally important to review the current treatments available for mild to moderate COVID-19 (mCOVID19), most notably the novel oral antivirals paxlovid and molnupiravir.

## 2. Methods

### 2.1. Search Strategy for COVID-19 Outcomes in Cancer Patients

The main online database, PubMed, was searched from inception until March 2022 for studies on cancer and COVID-19 outcomes, and trials evaluating treatments against mild-to-moderate COVID-19. Two different query strings were used for the cancer and COVID-19 outcomes search: (“Observational Study” [Publication Type]) AND (((lung neoplasm) OR (lung cancer)) OR (“Lung Neoplasms”[Mesh])) AND ((((COVID-19) OR (SARS-CoV-2)) OR (“COVID-19”[Mesh])) OR (“SARS-CoV-2”[Mesh])))((((neoplasm) OR (cancer) OR (Malignancy) OR (“Neoplasms”[Mesh])) AND ((((COVID-19) OR (SARS-CoV-2)) OR (“COVID-19”[Mesh])) OR (“SARS-CoV-2”[Mesh])))) AND ((outcome) OR (determinant)) AND (cohort).


The first query was used to search specifically for observational studies pertaining to lung cancer and COVID-19 outcomes, while the second query was broadened to all cancers. Regarding the eligibility criteria, we focused on articles with titles or abstracts that (a) had patients with active cancer or recent cancer treatment at the time of acquiring COVID-19, either evaluating only lung or thoracic cancer patients or analyzing all cancer types but containing a substantial number of lung cancer patients, and of both sexes; (b) analyzed outcomes of COVID-19 severity or mortality; (c) original studies that were observational cohort studies; and (d) whether anticancer treatments affected COVID-19 outcomes was evaluated.

### 2.2. Search Strategy for COVID-19 Treatments

The following query string was used to search for studies pertaining to therapies used to treat mild-to-moderate COVID-19:

((“Drug Therapy”[MeSH Terms]) OR (drug therapy)) AND (((((COVID-19) OR (SARS-CoV-2)) OR (“COVID-19”[Mesh])) OR (“SARS-CoV-2”[Mesh]))).

The PICOS search tool was used to selecting relevant studies pertaining to mild-to-moderate COVID-19 treatments [12]: (1) population: patients with mild-to-moderate COVID-19 not needing hospitalization; (2) intervention: drug or treatment in oral or intravenous forms given within 7 days of symptom onset that can decrease the SARS-CoV-2 viral load and hasten recovery; (3) comparison: the best standard of care available at the time of the clinical trial; (4) outcomes: rate of hospitalization 14 to 30 days after SARS-CoV-2 infection; (5) study: phase III randomized controlled clinical trials. We focused on phase III clinical trials with significant findings of treatment efficacy. Phase III clinical trials respecting the eligibility criteria set not found in the database search were added by the authors later during the review and editing process.

## 3. Pathophysiology of COVID-19

### 3.1. SARS-CoV-2 Invasion into Host Respiratory Cells

COVID-19 is caused by the single-stranded RNA virus SARS-CoV-2, which primarily spreads by airborne transmission [13]. COVID-19 mainly affects the upper respiratory tract but can extend to the lower respiratory tract and cause potentially lethal pneumonia [14,15]. SARS-CoV-2 can also affect a wide range of other systems, including the cardiovascular, hematological, and gastrointestinal tract systems. Since the virus mainly causes respiratory disease and LC is the main cancer type discussed, respiratory COVID-19 will be the focus of this review [16]. Regarding the pathogenesis of the SARS-CoV-2 infection, COVID-19 is described as a two-phase disease, with a viral replication phase and an overreactive, pathological immune phase [17]. SARS-CoV-2 entry is initialized through cleavage and activation of viral spike (S) proteins by host furin-like enzymes, followed by S protein attachment through its receptor binding domain (RBD) to angiotensin-converting enzyme-2 (ACE2) receptors on respiratory epithelial cells [18,19]. Infection by SARS-CoV-2 through ACE2 occurs in synergy with the host’s transmembrane serine protease 2 (TMPRSS2). More specifically, the S protein is cleaved by TMPRSS2, exposing viral membrane proteins that bind to the cell membrane and facilitate virus–cell fusion [20]. This virus–host interaction leads to viral entry and completion of the viral replication cycle, with subsequent transmission through viral shedding. During this initial phase, infected individuals are often asymptomatic due to a minimal immune response but are highly infectious as the virus replicates and spreads in the conducting airways. Subsequent viral extension occurs within the upper respiratory tract, leading to a symptomatic immune response with common cold symptoms of dry cough, fever, and generalized malaise. Most COVID-19 infections resolve and do not progress past this stage, as the immune system can adequately suppress viral replication and contain the infection [21].

### 3.2. Pathological Immune Response to SARS-CoV-2 Infection

The second phase, seen only in cases of sCOVID-19 pneumonia, is the result of an overreactive and generalized immune response to a viral infection of the lower respiratory tract, specifically the lungs. ARDS can develop in this phase, leading to acute respiratory failure and death. ARDS is a form of severe lung injury, clinically diagnosed by severe hypoxemia and bilateral radiographic opacities following SARS-CoV-2 infection and is marked by damage in the lung parenchyma, increased pulmonary vascular permeability, and a considerable loss of oxygenated lung tissue. ARDS is further characterized by extensive pulmonary vascular injury, with endothelial dysfunction, including upregulated pro-coagulation factors, inhibited fibrinolysis, and pro-thrombotic coagulopathies [22,23]. In ARDS, a dramatic increase in systemic pro-inflammatory cytokines is observed in sCOVID-19, known as CRS, causing inflammatory tissue damage [24]. The pro-inflammatory cytokine interleukin-6 (IL-6) is produced normally during infection or tissue damage and activates a cascade of pro-inflammatory signaling events during initial immune responses against pathogens but becomes overexpressed in CRS. Persistent expression of IL-6 in CRS is associated with poor viral clearance and immune function, leading to a pro-inflammatory lung microenvironment with poor specific virus clearance that causes extensive lung injury and ARDS [25]. Meta-analyses have further confirmed elevated levels of IL-6 and CRP, a marker of systemic inflammation, being consistently associated with severe ARDS and mortality along with increased C-reactive proteins (CRP) [26,27,28,29]. Lymphopenia has also been observed to occur in sCOVID-19 and suggests poor targeted immune function and viral clearance during the later stages of infection [29,30,31]. T cells from sCOVID-19 patients showed increased levels of the exhaustion markers PD-1 and Tim-3, and the immune-inhibiting cytokine IL-10, indicating poor T cell effector function [29,32,33]. Overstimulated T cells from persistent infection can eventually become functionally exhausted, further delaying viral clearance, and exacerbating inflammatory lung injury through the release of more pro-inflammatory cytokines.

### 3.3. The Physiological Effects of Comorbidities on SARS-CoV-2 Infection and Morbidity

Given the viral S protein–host ACE-2 receptor interaction for initial viral entry, comorbidities that affect host ACE-2 receptor levels may lead to increased rates of SARS-CoV-2 infection and subsequent severe disease due to altered immune statuses. For example, type II diabetes mellitus (T2DM) has been found to be causally related to an increased level of ACE-2 receptors, along with increased furin-like enzymes needed for S protein activation [34]. Diabetic individuals may have an increased rate of infection given potentially easier viral entry. Additionally, an altered immune status has been found in diabetic murine models, showing fewer levels of macrophages and CD4+ T cells; diabetic patients, thus, could have a higher risk of sCOVID-19 due to an altered immune system [35].

The impact of other comorbidities on the risk of COVID-19 infection and severity have also been described: hypertension, in which ACE-2 inhibitors as treatment for hypertension are known to upregulate ACE-2 receptor expression, leading to higher susceptibility to COVID-19 infection; COPD, where upregulated ACE-2 receptor expression, lung damage, altered local immunity, and increased mucous production contribute to increased COVID-19 infection and morbidity; and obesity, which is linked to reduced oxygen saturation in the blood due to compromised ventilation at lung bases and the presence of low-grade chronic inflammation, with abnormal levels of cytokines, adipokines, and interferon expression due to an altered immune system, making obese individuals susceptible to COVID-19 [34]. Potential for increased COVID-19 morbidity and mortality in diabetes, hypertension, COPD, and obesity have been further validated by meta-analyses evaluating these co-morbidities and COVID-19 outcomes [36,37].

LC patients, and generally many cancer patients, are older, with multiple co-morbidities, making them highly susceptible to SARS-CoV-2 infection and severe disease. A study by Fowler et al. [38] determined a high prevalence of comorbid diseases, including hypertension, COPD, and T2DM, in cancer patients, with the highest prevalence found in LC. Around 20% and 25% of LC patients had hypertension and COPD, respectively. These patients have an increased likelihood of having concurrent comorbidities, with COPD being commonly seen in combination with T2DM, cardiovascular disease, and congestive heart failure [38]. In addition to co-morbidities affecting COVID-19 outcomes in cancer patients, the pathophysiology of lung cancer could have an independent role in increasing the risk of sCOVID-19, which will be discussed.

## 4. Pathophysiology of Lung Cancer and Its Similarities with Severe COVID-19

### 4.1. Epidemiology, Diagnosis, and Pathology of Lung Cancer

LC is the second most commonly diagnosed carcinoma worldwide, in both men and women, with an estimated 234,030 new cases occurring in the U.S. alone in 2018 [39]. LC, being heavily associated with cigarette smoking, has seen a decrease in incidence rates and deaths since the 1980s in industrialized countries that have instituted tobacco control policies. However, LC incidence rates worldwide have been increasing due to the high prevalence of smoking in other countries, such as China and Brazil [40]. Screening for LC with low-dose computed tomography is currently recommended by the U.S. Preventive Services Task Force in high-risk patients. The indications are current smokers or former smokers who quit within the past 15 years, aged between 55 and 80 years old, and a smoking history of at least 30 pack-years. LC often presents symptomatically at diagnosis, with the most common respiratory symptoms of cough, dyspnea, and hemoptysis, and systemic symptoms of weight loss and anorexia. Digital clubbing is another common finding in LC. Clinical suspicion of LC is initially evaluated with chest radiography and is followed by contrast-enhanced computed tomography and positron emission tomography if suspicion remains high. Tumor biopsy is the next step for histopathological typing and molecular testing, as it can confirm diagnosis and plan treatment [41].

Lung cancers are classified based on their histopathological appearances into non-small cell lung cancers (NSCLCs) and small cell lung cancers (SCLCs). NSCLCs occur in 80% of all LCs and are further subdivided into multiple subtypes, including adenocarcinomas, squamous cell carcinomas, and large cell carcinomas, accounting for approximately 40%, 20%, and <3% of all LCs, respectively. SCLCs account for the remaining 20% of LCs and are almost exclusively associated with smoking. They are often highly aggressive with evidence of metastatic disease already present at the time of diagnosis. Patients with SCLCs are highly responsive to treatment, but most relapse within two years following treatment. Microscopically, the sizes of SCLC tumor cells are much smaller versus NSCLCs, with a fine granular chromatin without prominent nucleoli, a high mitotic and apoptotic rate, leading to necrosis, and a scanty cytoplasm. SCLCs are often located centrally, along the major airways [42].

Trends in overall LC mortality rates have decreased recently due to significant advances in clinical management of LCs (e.g., immunotherapy, targeted therapies, stereotactic body radiation therapy, etc.) [43]. However, mortality from LC remains high, accounting for 25% of annual cancer fatalities in the U.S. [40]. Data extrapolated from the Surveillance, Epidemiology, and End Results program (the SEER program) databases between 2012 and 2018 showed an overall five-year survival rate of LC at 22.9%, but the five-year survival rate greatly varies depending on tumor type and staging. The average five-year survival rate in localized stage disease is 61.2%, while advanced-stage cancers have a survival rate of 33.5% and metastatic lung cancer has a 7.0% five-year survival rate [44].

To provide more insights into the relationship between SARS-CoV-2 and LC, we will discuss the pathophysiological roles of increased inflammation and immunosuppression in LC.

### 4.2. Inflammation and Impaired Immunity in Lung Cancer

In LC, inflammation is known to play a pivotal role in tumorigenesis and cancer progression. A chronic inflammatory state induced by smoking or chronic infections is consistently associated with LC risk [45,46,47,48,49]. Prolonged pulmonary inflammation is further demonstrated in a meta-analysis, wherein a chronic CRP increase was associated with LC incidence [50]. During this stage of chronic inflammation, certain pro-inflammatory mediators are consistently produced and secreted into the tumor microenvironment (TME). IL-6 is suggested to be intimately implicated during the course of LC as a pro-inflammatory modulator of the TME, as demonstrated in many stages of LC pathogenesis. IL-6 is shown to be elevated in the bronchoalveolar lavage fluid of LC patients [50]. Consistently high levels of IL-6 and, thus, chronic inflammation, are also associated with an increased risk of developing LC [51]. High-serum IL-6 is associated with worse prognosis in non-small cell LC (NSCLC) patients, as increased IL-6 has been correlated with a lower 24-month survival rate and more advanced tumor staging [52]. It has also been demonstrated that IL-6 can promote metastasis of LC, as IL-6 produced by cancer-cell-activated astrocytes can promote further tumor cell proliferation in the brain of murine models [53]. Immunosenescence is another important hallmark in LC. T cells that target and suppress cancer cells can become functionally exhausted in cancer due to persistent stimulation by pro-inflammatory cytokines in the TME [54]. Immunosuppression is further marked by tumor cells being able to specifically target and deactivate cancer-targeting T cells through increased CTLA-4 and PD-1 signaling, cellular exhaustion markers functioning as inhibitors of T cell activation, proliferation, and survival, leading to immunosenescence. Clinically, increased expression of the PD-1 ligand on cancer cells is associated with significantly worse prognosis in LC patients [55]. These findings suggest that inhibition of the T cell effector function enhances tumor cell immune evasion and survival, leading to cancer progression.

While COVID-19 ARDS is acute and LC pathogenesis is chronic, aberrant inflammation and immune dysregulation appear to play important roles in LC and sCOVID-19. A heightened inflammatory state marked by elevated IL-6 and CRP are involved in progression and poor outcomes in both diseases. Regarding altered immunity, significantly increased levels of PD-1 are seen in both sCOVID-19 and LC. This increase in T-cell exhaustion markers suggests that suppression of T-cell-mediated immune function occurs through PD-1/PD-1L interactions and plays an important role in both diseases; immune cells cannot effectively clear their targets, resulting in either viral persistence or LC tumor survival. The immunosuppressed and chronic pro-inflammatory TME in LC patients may provide an optimal environment for SARS-CoV-2 replication while simultaneously increasing the risk to develop the aberrant inflammatory response seen in COVID-19 CRS.

### 4.3. Aging-Related Changes Associated with Lung Cancer and Severe COVID-19

Age is associated with both LC incidence and sCOVID-19 [8,40]. Significant changes in inflammatory and immunological functions are observed with aging, with increased age-related low-grade chronic inflammation, known as “inflamma-aging”, and generalized decreased immune function, known as immunosenescence. These age-related changes could be important factors increasing the risks of both LC and sCOVID-19. Age-related increased NF-κB signaling is known to activate multiple pro-inflammatory cytokines and mediators, including TNF-α/β, IL-6, and CRP [56]. This activation of inflammatory pathways is reflected in increased blood concentrations of TNF-α, IL-6, and CRP in aging individuals [57]. Increases in inflammatory mediators related to age are hypothesized to be caused by an increase in disease-associated molecular patterns (DAMPs) and pathogen-associated molecular patterns (PAMPs). The rise in DAMPs has been associated with age-related decrease in macrophage-dependent cell clearing of damaged, senescent, and necrotic cells, while increased PAMPs may be related to the reactivation of latent infectious diseases acquired over a lifetime [58].

Additionally, effective innate immune responses become compromised with aging and can lead to an increase in generalized inflammation, as perhaps a compensatory immune mechanism. Type I interferon (IFN) production, a signaling protein essential for initial antiviral response, appears to be delayed with aging, leading to increased pro-inflammatory cytokine release and a decrease in anti-viral natural killer cell activation. Innate immune cells also have a reduced capacity to respond effectively to DAMPs and PAMPs. Neutrophils migrate less accurately during infection and have reduced phagocytic and intracellular killing capacities, while macrophages that secrete pro-inflammatory cytokines are more easily activated [58]. In the lungs, pulmonary dendritic cells (pDC) are shown to migrate less efficiently to regional lymph nodes for antigen presentation. Reduced virus-specific cytokine production in pDCs is also observed and is highly correlated with poorer antibody responses to viral infections. General intracellular cytokines in pDCs are, conversely, shown to be increased with aging, contributing to a pro-inflammatory TME [59]. Less effective anti-viral and anti-cancer responses with more generalized inflammation occur because of these age-related changes in innate immunity.

T cells are required to generate a highly effective and specific immune response against invading pathogens. In the context of aging and immunosenescence, the naïve T-lymphocyte population dramatically decreases in older adults due to involution of the thymus, the main site of T-cell production during puberty, and incurs a further decrease due to gradual, continuous age-related hematopoietic stem cell insufficiency [60]. CD8+ T cells, crucial in targeting virally infected and neoplastic cells, have increased expression levels of apoptotic PD-1 and CTLA-4 signaling in aging murine models, suggesting decreased T-cell effector function related to aging [61]. Simultaneously, a continuous increased expression of proinflammatory mediators (e.g., TNF-α, IL-2, IL-6, and IFN produced by CD8+ T cells) has been associated with increasing age [57]. As a result, T-cell-dependent immune responses to new antigens from both emerging pathogens, such as SARS-CoV-2 and tumor cells, may become compromised with aging, with T cells playing a potentially significant role in an increased age-related inflammatory state.

Because of inflamma-aging and immunosenescence, both antiviral and anticancer immune functions weaken in aging adults, with LC patients, who are often older, expected to be particularly vulnerable to sCOVID-19. In COVID-19, a low-grade chronic inflammatory state in aging adults provides favorable conditions for the progression of COVID-19 to CRS and ARDS. Meanwhile, impaired innate and adaptive immune responses lead to viral persistence, which further exacerbates inflammation. In LC, an altered lung TME from inflamma-aging and decreased immunosurveillance are able to elicit tumor progression [62].

## 5. Impact of Lung Cancer on COVID-19 Outcomes

Cohort studies measuring COVID-19 outcomes in cancer patients have shown that certain cancer types suffered from higher rates of sCOVID-19 versus the general population. An early Italian study reported that patients with hematological cancer or LC were more vulnerable to sCOVID-19 than other cancers. Of note, LC was associated with a fourfold increased risk of mortality from sCOVID-19 versus non-cancer COVID-19 patients [63]. A second early cohort study evaluating 102 patients with LC and COVID-19 was conducted in the U.S. The report showed that SARS-CoV-2 infection in LC was longer and more severe than the general population, with 62.0% hospitalized, 21.0% admitted in the ICU, 18.0% receiving intubation, and a 25.0% mortality rate [64].

### 5.1. Characteristics of Lung Cancer Patients Associated with Severe COVID-19

Given that early studies reported increased rates of sCOVID-19 and mortality in LC, we reviewed larger cohort studies to clarify the characteristics of general cancer patients and LC patients with COVID-19 to identify the risk factors for sCOVID-19 in this patient group. A summary of the studies discussed below recapitulates the cohort sample size, COVID-19 outcomes, and risk factors associated with sCOVID-19 (Table 1).

Overall, COVID-19 mortality in cancers patients ranges from 13% to 33.7%, rates that appear much higher compared to non-cancer COVID-19 patients (Table 1). Regarding the CCC19 study, a later report by the same group that included more patients identified similar risk factors and additionally discovered a 26.0% mortality rate in patients with thoracic cancers, higher than both hematological malignancies (17.0%) and overall solid tumors (13.0%) [65]. Indicators for sCOVID-19 were also much higher for cancer patients with COVID-19 versus non-cancer patients. For example, hospitalization rates for patients who recently received cancer treatment was 33.7%, more than double the rate of 14.6% for non-cancer patients [66]. The Sharafeldin et al. study also showed that hospitalized patients with more severe disease were higher in the cancer patient group versus the non-cancer group; 8.2% of hospitalized cancer patients required mechanical ventilation versus 5.2% in non-cancer patients [67].

Hospitalization rates for cancer patients ranged as high as 76% and 78.3% in the studies by Provencio et al. [9] and Garassino et al. [10], respectively. Of note, however, these studies included a high proportion of patients with advanced cancers; 74% of patients had stage IV cancers, with the overall most frequent cancer type non-small cell lung carcinomas (76%) in the Garassino et al. [10] paper. Further, 79.2% of patients in the Provencio et al. [9] study had either advanced stage III, metastatic, or unresectable malignancies, with the most frequent cancer type being non-small cell carcinomas, at 76%. Patients in the Garassino and Provencio studies with advanced respiratory diseases likely had several risk factors for sCOVID-19, which will be discussed and, thus, a poorer health status that made these patients more vulnerable to sCOVID-19.

In the studies reviewed, the most frequent risk factors for COVID-19 identified in cancer patients were older age, co-morbidities (e.g., COPD, T2DM, etc.), male sex, and smoking history (Table 1). Certain cancer types were also identified to be associated with sCOVID-19 outcomes, including hematological and respiratory malignancies in the De Joode et al. and the Chavez-MacGregor et al. studies [66,68]. Finally, advanced cancers and metastasis were frequently associated with increased sCOVID-19 and mortality in the study of Provencio et al. [9].

**Table 1 cancers-14-03598-t001:** Summary of studies evaluating COVID-19 outcomes in cancer patients.

Author, Year	Number of Patients in Study	Outcomes (All Values in %)	Risk Factors for sCOVID-19
Kuderer et al. [69]	928 cancer patients	13 mortality cancer vs. 1.6 non-cancer	Older age, co-morbidities, and smoking.
Assaad et al. [70]	425 cancer patients	27.8 mortality COVID-19 (+) cancer patients vs. 16.3 COVID-19 (−) cancer patients	Male, lymphopenia, T2DM, immunosuppressive treatment, and metastasis.
Sharafeldin et al. [67]	63,413 cancer patients	14.8 all-cause mortality, and 8.2 requiring mechanical ventilation in COVID-19 (+) cancer patients vs. 12.5 and 5.2, respectively, in COVID-19 (−) cancer patients.	Older age, male, co-morbidities, and multisite tumors.
De Joode et al., 2020 [68]	442 cancer patients	32.3 mortality	Older age, male sex, prior or concurrent cancers, lung cancer, and hematological malignancies.
Ferrari et al. [71]	198 cancer patients	16.7 mortality	Older age, smoking, and advanced malignancies.
Chavez-MacGregor et al. [66]	493,020 COVID-19-positive and without cancer 9,991 cancer patients without recent cancer treatment 4,296 cancer patients with recent treatment	Hospitalization: 14.6 without cancer, 25.2 without recent cancer treatment, 33.7 with recent cancer Tx. Mortality rate: 1.6 without cancer, 5.0 without recent cancer treatment, 7.8 with recent cancer treatment	Older age, co-morbidities, male, Hispanic, African American, obesity, metastasis, lung cancer, and hematologic malignancies.
Garassino et al. [10]	200 thoracic cancer patients	26 mortality due to COVID-19 76 hospitalization 88 hospitalized meeting ICU criteria 10 ICU admission rate	Smoking history
Provencio et al. [9]	447 lung cancer patients	78.3 hospitalization rate 32.7 mortality 2.0 ICU admission rate	Older age, co-morbidities, smoking history, concomitant administration of NSAIDs, lymphopenia, high LDH, low albumin, and advanced malignancies.

Cancer patients appear to be at a high risk of mortality overall, with lung cancer being associated with increased COVID-19 mortality in two studies, including the large-scale study by Chavez-MacGregor et al. [66]. Furthermore, cancer patients also had more frequently severe COVID-19 with higher rates of hospitalization and mechanical ventilation versus non-cancer patients. Risk factors associated with worsened COVID-19 outcomes in cancer patients were older age, co-morbidities, smoking history, and the male sex. Other potential risk factors for sCOVID-19 related to anticancer therapies were also evaluated in the studies reviewed and will be discussed next.

### 5.2. Impact of Lung Cancer Treatments on COVID-19 Outcomes

Previous reports showed, early in the pandemic, concerning data about their administration, systemic anti-cancer therapies as a risk factor for increased COVID-19 morbidity and mortality [72,73]. Regarding anticancer therapy and sCOVID-19 risk, relevant information from all the cohort studies discussed above is summarized in Table 2. No significant association was found between poor COVID-19 outcomes and the following therapies: immunotherapy, targeted therapy, hormonal therapy, and surgical resection (Table 2); however, RT and sCT showed conflicting results. The N3C [67] and Chavez-MacGregor et al. [66] studies observed significantly increased mortality in COVID-19 patients with recent sCT treatment compared to all other studies discussed (Table 2) [9,10,66,67,68,69,70,71,74]. sCT is generally given to patients with more advanced or metastatic tumors and worse prognoses, which are associated with increased risk of sCOVID-19. Additionally, the N3C [67] and Chavez-MacGregor et al. [66] studies evaluated cancer patients with solid and hematological malignancies. One meta-analysis showed that recent sCT treatment increased the risk of COVID-19 mortality only in patients with hematological malignancies compared to solid tumor malignancies [75]. This difference could be explained by the additive myelosuppressive effects of both hematological cancers and sCT, which significantly increase the risk of lymphopenia. Lymphopenia has been identified as a major independent risk factor for sCOVID-19, as it could prevent adequate immune function against SARS-CoV-2 [76,77]. Regarding RT, while other studies did not find an association between RT and greater COVID-19 mortality, Chavez-MacGregor et al. identified a significant association between sCOVID-19 and recent RT [66]. RT has also been noted to induce lymphopenia, with patients having advanced NSCLCs and receiving high-RT doses shown to be at increased risk for lymphopenia [78,79].

Cancer therapies have been affected by COVID-19 in ways other than increased risk of sCOVID-19 outcomes. Certain LC therapies might account for complications with clinical presentations similar to COVID-19 pneumonia. While no association was found between immunotherapy and COVID-19 mortality, the administration of immune checkpoint inhibitors (ICIs) as immunotherapy can lead to severe pneumonitis as a complication. Checkpoint-inhibitor-induced pneumonitis (CIP) is caused by inflammation in the lung due to uncontrolled reactivation of T-cell activity and reversal of exhaustion signaling, with an incidence rate of 3–5% in cancer patients undergoing ICI treatment [80]. Clinically, CIP presents with cough, dyspnea, and sometimes fever. Radiological findings for CIP include patchy peripheral consolidation, ground-glass opacities (GGOs), interstitial and interseptal lobular thickening, and centrilobular nodules [81,82]. Similarly, radiation-induced pneumonitis (RIP) presents with cough, dyspnea, and fever, with radiological findings that include GGOs and patchy consolidations [83]. CIP and RIP findings closely resemble COVID-19-associated pneumonia, which also clinically presents with cough, dyspnea, and fever, along with GGOs and consolidations [16]. Therefore, accurate diagnosis of COVID-19-associated pneumonia might be challenging for LC patients receiving recent ICI or RT. Misdiagnosis of COVID-19 pneumonia with RIP or CIP and vice versa can have drastic consequences. Corticosteroid use in COVID-19 only showed clinical benefits in patients with ARDS and requiring supplemental oxygen, where they depend on the correct timing and dosage; wrongful use can lead to decreased immune function and viral clearance, increased disease severity, and higher mortality [84,85,86,87,88,89]. Conversely, CIP and RIP can be life-threatening complications that require early detection and treatment. Misdiagnoses can lead to costly delays and irreversible lung damage [83,90]. A summary of the similarities and differences between CIP, RIP, and COVID-19 pneumonia is described in Table 3.

## 6. Cancer Care during the COVID-19 Pandemic

### 6.1. Pandemic-Related Adaptations in Care Delivery and the Use of Specialized COVID-19-Free Oncology Hubs

Healthcare settings worldwide have been immensely affected by the high volume of patients due to the pandemic, combined with an increased need for intensive-care beds and iatrogenic transmission of SARS-CoV-2. As a result, hospitals were forced to deliver alternative caregiving models. Otolaryngology (ENT) departments are especially vulnerable to SARS-CoV-2 transmission given their frequent direct contact to saliva and mucous suspensions, ranging from physical examination of the oral cavities to procedures, such as rhinoscopies and laryngoscopies. ENT clinics implemented strict alternative standard operating procedures to minimize contact to SARS-CoV-2 during the pandemic. Individual safety practices in clinics include COVID-19 screening before ENT evaluation, the use of telemedicine in non-urgent evaluations, and the use of PPE (e.g., disposable gloves, eyewear, FFP2-grade masks, gowns, etc.). Changes to the physical environment were also implemented, involving the use of well-defined segregated areas for structurally isolated ENT outpatient rooms, a filter room with well-defined segregated outpatient rooms, filter areas where patients enter wearing a face mask, and a dirty area for safe removal of PPE with a separate clean area dedicated to PPE wearing. Similar changes were implemented in operation rooms as well [91]. In cancer care settings, similar rules should be followed given the vulnerable patient population to sCOVID-19 and the need to protect healthcare providers from SARS-CoV-2 exposure.

The use of PPE, specifically face masks, is crucial in protecting healthcare workers from SARS-CoV-2. Long-term and constant use of face masks does have its drawbacks; however, a survey study by Maniaci et al. [92] described a significantly higher prevalence of respiratory and ocular symptoms, including nasal symptoms, itchiness, and redness of the eyes, coughing, and difficulty breathing, with continuous mask wearing. These mask-related symptoms are worsened by co-morbidities affecting the respiratory tract (e.g., chronic obstructive pulmonary disease (COPD), allergic rhinitis, etc.) and have led to a perceived decrease in work performance and quality of life amongst healthcare workers. Given the importance of mask wearing, the authors implemented several recommendations, including smoking cessation, use of nasal wash, air humidification, and treatment of co-morbidities, which have significantly decreased mask-related symptoms [92]. These face-mask-related effects can be extended to any occupation requiring continuous mask wearing and the recommendations elicited by Maniaci et al. [92] should be applied.

The circumstances of the COVID-19 pandemic drastically altered cancer care. Throughout the pandemic, healthcare staff implemented many measures to minimize SARS-CoV-2 exposure and continue cancer care. A strategy to prevent COVID-19 outbreaks was used in an Italian radiation oncology hub designed to pursue regular cancer treatment while minimizing COVID-19 exposure during the initial months of the pandemic. The European Institute of Oncology (IEO) received all regional cancer patients from general hospitals and established a central COVID-19-free hub, using strict protocols effective in protecting cancer patients from COVID-19 infection. COVID-19 diagnosis, isolation, and management were rapid as staff had protocols to identify suspicious symptoms (e.g., fever, cough, dyspnea, diarrhea, etc.) and wide access to same-day diagnostic tests. Additionally, the center minimized the risk of exposure with telemedicine, social distancing, remote working, requirements to wear surgical masks for everyone and PPE for staff, prohibiting relatives and friends from visiting, and mandatory nasopharyngeal swabs for incoming patients [93]. Similar measures were implemented in the Italian University Hospital of Bologna. Its gynecological oncology department followed similar protocols during the initial lockdown period of the pandemic and was able to prevent outbreaks from occurring, while maintaining pre-pandemic volumes of cancer care [94]. Thus, COVID-19-free oncological hubs specialized in continuing regular cancer care are a viable strategy during COVID-19 outbreaks.

### 6.2. Changes in Lung Cancer Treatment Delivery

Certain cancer treatments cannot be delayed, even during a pandemic, due to the rapidly progressive nature of lung neoplasms. Anticancer regimens have been altered to maintain cancer care while minimizing the risks of iatrogenic SARS-CoV-2 exposure and infection. Durvalumab, a PD-1-PD-1L inhibitor, is used as adjuvant immunotherapy in stage III NSCLC. While durvalumab was normally given bi-weekly at 10 mg/kg, the regimen was altered to 20 mg/kg every 4 weeks to reduce hospital visits and minimize contact with SARS-CoV-2, which was complemented with telemedicine for safety monitoring of patients. Despite the doubling of dosage per visit, a study by Joshi et al. [95] showed a better safety profile for the 4-week regimen compared to the 2-week regimen (toxicity rate of 7.0% versus 15.0%, respectively). The study has important implications for the safety of the altered regimen. The longer interval between durvalumab doses allowed the possibility to minimize SARS-CoV-2 exposure while maintaining cancer care during COVID-19. Additionally, this alternative dosing modality also demonstrated clinical benefit by decreasing treatment-related toxicity, which is potentially significant in providing better cancer care [95].

The use of shortened high-dose hypofractionated RT regimens (HFRT) was suggested as an alternative treatment modality to conventionally fractionated RT (CFRT) during the pandemic [96,97]. Indeed, a high-dose hypofractionated RT regimen can be completed in a shorter time with less hospital visits and less SARS-CoV-2 exposure. Currently, CFRT remains the preferred regimen in treating most LCs; the efficacy and safety of HFRT are not fully understood. A phase II study by Cheung et al. [98] showed, in 80 patients, that HFRT, delivering 60 Gy in 15 fractions, led to favorable primary tumor control rates (87.4% at two years) and survival rates (68.7% at two years) in stages T1–3 N0 M0 NSCLC patients, with severe toxicities being uncommon with HFRT treatment [98]. Iocolano et al. also observed no significant difference in overall survival with HFRT versus CFRT [99]. However, large clinical trials investigating whether HFRT is equal or superior to CFRT have not been conducted. While a minority of RT departments currently recommend HFRT for LCs, HFRT recommendations for LC, when CFRT cannot be reasonably delivered, are significantly more common versus other cancers (e.g., cervical and head and neck cancers); this difference could be attributed to the higher risk of sCOVID-19 in LC patients. Additionally, if a SARS-CoV-2 outbreak occurs or health resources become limited, there is a strong consensus for switching to HFRT due to its benefits of minimal exposure from hospital visits [97]. The increased risk of adverse effects and potential RIP due to the increased doses could become problematic if HFRT is used as an alternative, however. In the case of switching to HFRT, there was an additional consensus on sequential chemoradiotherapy (sCRT) in treating stage III NSCLC to minimize prolonged hospital exposure to COVID-19 during treatment and the immunosuppressive effects of concurrent CRT (cCRT) [100,101]. A phase III clinical trial evaluating sCRT versus cCRT in stage III NSCLC patients showed decreased short-term toxicity in sCT versus cCRT. However, the treatment response and five-year survival rates in sCRT were inferior to cCRT, which may impact LC outcomes if sCRT was used as an alternative treatment during the pandemic [102].

Surgical services are especially vulnerable to delays due to the high risk of SARS-CoV-2 exposure during surgery-related hospitalization and the redirection of surgical resources (e.g., mechanical ventilators, anesthesiologic services, and ICU beds) for COVID-19 patients during outbreaks. Resectable LCs (e.g., stages I/II NSCLC), thus, require alternative treatments when timely surgery cannot be performed. Stereotactic body radiotherapy (SBRT) for resectable stage I/II NSCLC has been suggested as a viable alternative [100]. SBRT requires fewer resources and can be performed in 3–5 fractions, without requiring hospitalization. However, surgery remains the standard of care for resectable NSCLCs [103,104]. Patients with resectable NSCLCs, treated with only SBRT during COVID-19, may be at risk of sub-optimal cancer care. To address this potential gap in care, a recent review proposed a 3-month follow-up surgical resection after initial SBRT (the SABR-BRIDGE approach), once local COVID-19 prevalence has decreased to acceptable levels and elective surgeries have resumed. The proposed SABR-BRIDGE method can, thus, provide timely care to LC patients during the pandemic, while minimizing the potential risk of cancer recurrence following SBRT through surgical means [105].

## 7. Impact of COVID-19 on Lung Cancer Screening, Diagnosis, and Treatment

### 7.1. Decreased Patient Volumes and Delays in Cancer Care Due to COVID-19

Despite the current pandemic adaptations from cancer centers, initial delays in cancer diagnoses and treatment were paramount to minimize infection given the risk of sCOVID-19 in LC patients. The early consensus in the CHEST panel report was to delay all screenings until further notice, delay surveillance procedures and evaluations of suspicious nodules for three to six months, and only treat confirmed stage I NSCLC without any delay [106]. These delays have been widely implemented with COVID-19 restrictions, with local health authorities ceasing all non-essential clinical activities and ramping down surgeries in order to minimize COVID-19 exposure and allocate resources to combat the pandemic [107]. The initial objective was to wait until the local prevalence of COVID-19 decreased to acceptable levels, where regular cancer care can then resume with minimal SARS-CoV-2 infection risk. A report recently modeled the impact of a COVID-19-related three-month delay of surgical resection and identification of suspicious lung nodules while considering risks of NSCLC progression, COVID-19 infection, and overall 5-year survival. The authors predicted that immediate resection led to slightly improved 5-year survival outcomes versus a three-month delay, but a delay led to better survival rates if the perioperative risk for SARS-CoV-2 infection exceeded 13.0% [108].

As COVID-19 continued with further outbreaks and waves, it became clearer that delaying cancer treatment was associated with high risks of cancer progression and mortality. Due to the delays that often exceeded three months due to high local prevalence and a general fear of visiting hospitals during the COVID-19 pandemic, a significant decrease in cancer diagnoses was detected during the first wave of the pandemic compared to before the pandemic [109]. An Ohioan university system reported that low-dose CT screenings for LCs were suspended for multiple months, while no-show rates for screening appointments increased by 25.0% during COVID-19 versus pre-COVID-19 (respectively, 15.0% vs. 40.0%; *p* < 0.04) [110]. These reductions can be observed throughout the U.S., which saw a 75.0% decrease in LC screenings and a 23.0% decrease in LC diagnoses in April 2020 [111,112]. Cancer screening tests in Ontario, Canada, similarly decreased by over 40.0% in 2020, with the largest decreases seen in May 2020, at over 90.0%, and in the UK, urgent referrals for cancer diagnoses fell by 70.0% [107,113]. Similar decreases can be observed in cancer treatments. sCT attendance for cancer care, in general, had fallen by over 40.0% in the U.K. at the outset of the pandemic. In Canada, 54.0% of cancer patients surveyed in the Canadian Cancer Survivor Network saw their cancer care appointments cancelled or delayed [113,114]. In the U.S., significant reductions in the billing of oncology products (−26.0%), sCT administration in hospitals (−21.0%), and cancer biopsies (−58.0%) were also observed, and these reductions in care lasted for multiple months [112]. Similarly, RT was affected worldwide at the beginning of the pandemic, where a majority of radiation oncology clinics surveyed internationally had a significant decrease in treatment in 2020, with average treatment volumes reduced to 68.0% in the U.S., 75.0% in Europe, and 59.0% in Latin America versus pre-pandemic treatment volumes [115,116].

### 7.2. Consequences of Delays in Lung Cancer Treatment

Worryingly, the decreases in screening, diagnoses, and treatment of cancers seen during the COVID-19 pandemic can potentially lead to a surge in cancer progression. The impacts of delays, interruptions, and cancellations of cancer care services at the initial stages of the pandemic have already been observed by oncological centers worldwide. A majority of the European radiation oncology centers (71.0%), surveyed one year following the onset of the COVID-19 pandemic, reported patients presenting with more advanced diseases [117]. In the U.S., 66.0% of surveyed physicians noticed more advanced cancers during the COVID-19 pandemic [118]. Regarding LCs, Van Hanren et al. found a significant increase in suspicious lung nodules following the resumption of screening programs in their hospital systems, from 8.0% pre-pandemic to 29.0% during the pandemic [110]. A U.K. study by Lai et al. [113] estimated between 7,165 to 17,910 total excess deaths in cancer patients for one year, with increased rRRs of 1.2–1.5 due to COVID-19. This estimate includes deaths directly attributed to COVID-19 complications and indirect deaths from healthcare service delays. Notably, patients with LC were estimated to have the highest volume in excess deaths (between 1,305 to 3,262 deaths) from the 24 different cancers types analyzed [113]. Regarding surgeries, curative treatments can be delayed to an incurable stage; a six-month delay of surgical resection over a year can lead to 1,439 excess deaths. A six-month delay can also lead to a reduction in overall 5-year net survival of approximately 30% for age groups between 30 and 79 years old, with stages II and III NSCLCs [119]. A meta-analysis on pre-COVID-19 treatment delays showed that a delay in all cancer treatments (i.e., surgery, RT, and sCT) of only 4 weeks can lead to a significant increase in mortality in stage III NSCLCs [120]. Additionally, the delays during COVID-19 in screening and treatment have led to an enormous backlog for oncological care as services resumed. Backlogs in certain screenings and surgeries have been estimated to be resolved only after more than one year, ranging from 21 to 41 months [121,122,123]. COVID-19-pandemic-related excess cancer mortality, especially from LC, is expected to significantly increase for the next few years with the ongoing backlog of cancer care expected to further increase life-years lost.

## 8. Need of Treatments against Early COVID-19 Due to Novel Variants of Concern

### 8.1. Mutations in SARS-CoV-2 and Its Variants of Concern

As SARS-CoV-2 evolves into novel variants of concern (VOC) over time, it is important to review how the ease of transmission of the virus, its infectivity, antigenicity, viral disease severity, and virulence, are altered in VOCs, with the B.1.617.2 (Delta) and B.1.1.529 (Omicron) VOCs being examples. Mutations in the S protein are the principal mechanism of increased virulence, transmissibility, and immune evasion among the novel VOCs. In Delta, 17 mutations were discovered in the S protein, with notable L452R and T478K mutations on the RBD of the S protein. These substitutions increase binding affinity to ACE2 and decrease immune recognition, enhancing replication and immune evasion [124]. Another noted mutation (P681R) was discovered in the furin cleavage site of S protein, which demonstrated increased virus-cell fusogenicity, a marker of virulence, in murine models [125]. Omicron, which has rapidly overtaken Delta worldwide in incidence rates, carries 32 mutations on the S protein, with 15 mutations on the RBD and 3 on the furin cleavage site, decreasing antibody affinity while increasing viral entry. This unusually high level of mutations may explain the increased infectivity and immune evasion of Omicron, which is expected to be twice as contagious as Delta and over 10-times versus wild-type SARS-CoV-2 [126,127].

### 8.2. Vaccine-Related Waning Immunity and Viral Resistance

To minimize delays of cancer diagnosis and treatment, COVID-19 infection and complications must be minimized by reducing exposure, vaccinations, and early treatment in case of infection. Despite efforts to minimize COVID-19 exposure, risk of SARS-CoV-2 transmission can be difficult to control in clinical settings, especially during outbreaks of novel VOCs. While COVID-19 vaccines have initially shown high efficacy in preventing symptomatic infection, waning immunity against COVID-19 from vaccination has become an increasing challenge. A rapid and significant decrease in humoral response following two doses of the BNT162b2 vaccine has been observed over a six-month period [5]. A similar trend was observed with the mRNA-1273 vaccine; effectiveness against symptomatic infection waned after four months and less than 50.0% efficacy was observed at the seventh month, but vaccination still remained protective against sCOVID-19, as protection against severe disease remained high following two doses after seven months [128]. However, the multiple mutations in the S protein in Delta and Omicron have further decreased vaccine efficacy, as with all current mRNA vaccines. BNT162b2 showed reduced efficacy in Delta versus the Alpha VOC with one dose, at 30.7% and 48.7%, respectively. Vaccine efficacy against Delta was largely restored with two doses, at 88.0%, versus 93.7% for Alpha [129]. Vaccine efficacy in preventing symptomatic infection was further reduced in Omicron. Large, significant reductions in neutralizing antibody responses were found in BNT162b2 (33-fold), the viral vector recombinant vaccine ChAdOx1 (14-fold), and mRNA-1273 (74-fold) against Omicron following two doses versus Alpha in a study by Willett et al. [127]. Their vaccine efficacy modelling estimated a further reduction in Omicron compared to Delta, while a third dose only partially restored humoral response against Omicron [127].

### 8.3. Breakthrough Infections in Cancer Patients

With the novel VOCs able to escape immunity in healthy individuals, cancer patients appear to have an even higher risk of developing symptomatic infection that can progress to sCOVID-19, despite being fully vaccinated. An observational study that included cases between November 2020 and May 2021 demonstrated that most cancer patients with breakthrough COVID-19 were hospitalized (65%). Severe disease was observed in these hospitalized patients, with 16% ICU admission and 13% mortality rates. These findings were comparable to unvaccinated cancer patients, as the study found no significant differences in mortality between vaccinated and unvaccinated cancer patients (adjusted OR, 1.08; 95% CI, 0.41–2.82). A lack of significant difference was equally found in ICU and hospitalization rates. Patients with active and progressive cancers, hematological malignancies, or undergoing systemic sCT were overrepresented among the vaccinated group with sCOVID-19, suggesting increased vulnerability in these groups. The study further identified lymphopenia as a significant risk factor for mortality in the vaccinated cancer group [130]. As this study was conducted in a population mostly affected by Alpha and wild-type variants, breakthrough infections and subsequent severe disease are expected to be worsened by Delta, Omicron, and future VOCs. Breakthrough infections are expected to become an increasing challenge in oncological care with the appearance of novel and future VOCs. Thus, there is a great need for therapies to prevent hospitalization and ICU admission in cancer patients. Many treatments have been determined in large-scale clinical trials to have benefits in reducing sCOVID-19 in non-hospitalized patients presenting with early mCOVID-19 and having at least one sCOVID-19 risk factor. Table 4 presents a summary of published phase III studies of therapies for mCOVID-19.

## 9. Therapies against Early COVID-19 Infection in Cancer Patients

### 9.1. Direct-Acting Antivirals

Direct-acting antivirals (DAAs) specifically target and inhibit the viral replication mechanism to decrease the viral load. The novel oral COVID-19 antivirals molnupiravir and combination nirmatrelvir–ritonavir (or paxlovid) have both been approved to treat mCOVID-19.

#### 9.1.1. Molnupiravir

Molnupiravir, in its active form β-D-N^4^-hydroxycytidine triphosphate, is a nucleoside analog of cytidine or uridine. Viral RNA-dependent RNA polymerase (RdRP) erroneously incorporates NHC triphosphate, leading to mutations, genomic instability, and a reduction in viral replication [139]. The phase III clinical trial MOVe-OUT evaluated early administration of molnupiravir in non-hospitalized and unvaccinated COVID-19 patients. The study determined that molnupiravir demonstrated a modest reduction in sCOVID-19, with a 30.9% relative risk reduction (rRR) [132]. Another phase III trial, titled MOVe-AHEAD, is evaluating the use of molnupiravir as post-exposure prophylaxis (PoEP) against COVID-19 in adults living in the same household of a person with confirmed COVID-19 (ClinicalTrials.gov Identifier: NCT04939428).

#### 9.1.2. Paxlovid

In paxlovid, the nirmatrelvir component is an inhibitor of coronavirus 3C-like protease, an enzyme required to lyse and release viral proteins essential for viral RNA replication and proliferation [140]. Co-administration of the antiretroviral ritonavir with nirmatrelvir in paxlovid inhibits hepatic CYP3A4 metabolism and prolongs the active effects of nirmatrelvir. The phase III trial EPIC-HR evaluated paxlovid in unvaccinated and non-hospitalized patients. The study demonstrated an 88.9% rRR in hospitalization and mortality rates caused by COVID-19 with paxlovid treatment. The incidence of all-cause adverse events was similar for both paxlovid and placebo. Paxlovid showed a higher rate of treatment-related adverse events, which was attributed to milder symptoms of dysgeusia and diarrhea [131]. Both medications have shown efficacy in treating early COVID-19, with paxlovid being particularly effective in reducing severe outcomes (88.9% rRR for paxlovid vs. 30.9% rRR for molnupiravir).

#### 9.1.3. Remdesivir

The DAA remdesivir, in addition to its current use for hospitalized COVID-19 patients, has also been evaluated to treat mCOVID-19 in outpatients with at least one risk factor for sCOVID-19. Remdesivir is a prodrug that is metabolized into its alanine active form once inside the body to become a nucleotide analogue. Active remdesivir is then incorporated into viral RNA by viral RdRP to promote early chain termination during RNA synthesis, inhibiting replication [141]. A phase III trial using remdesivir in outpatients with COVID-19 demonstrated an rRR of 87.0% in hospitalization and mortality. The rates of adverse events were similar between treatment and placebo groups [133]. However, the need for intravenous access to administer remdesivir limits its practicality for early at-home treatment.

### 9.2. Repurposed Drugs

#### 9.2.1. Fluvoxamine

Regarding medications that have been repurposed for COVID-19, fluvoxamine, as a selective serotonin reuptake inhibitor (SSRI), has been evaluated as a potential treatment for non-hospitalized mCOVID-19. Fluvoxamine is known as an inhibitor of acid sphingomyelase (ASM), a host enzyme essential for the generation of ceramide, which is known to facilitate viral entry [142]. Sigma-1 receptor (S1R) is a host dependency factor that has been proposed to inhibit early SARS-CoV-2 replication by interfering with initial virus-induced host cell reprogramming, where fluvoxamine has demonstrated high affinity to S1R as a potent agonist [142]. The TOGETHER trial evaluated fluvoxamine in non-hospitalized COVID-19 patients with a primary endpoint defined as an ER retention for over six hours or transfer to a tertiary hospital due to COVID-19 within 28 days of randomization. The study showed the efficacy of fluvoxamine in reducing its primary endpoint, with a 32.0% rRR, but all-cause hospitalization rates remained non-significant between treatment and placebo groups (10% vs. 13%; 95% CI, 0.58–1.06). Serious adverse events were also lower in the treatment group versus placebo [134]. Nonetheless, the trial contains various key limitations, including a non-significant rate of actual all-cause hospitalization rates, and the fact that their primary endpoints of ER retention as proxy for hospitalization were not used in other well-established studies, questioning ER retention and hospital transfer as clinically relevant metrics. More phase III clinical trials are required to determine the efficacy of fluvoxamine. If effective, the oral fluvoxamine, along with its accessibility and affordability, could potentially be a highly advantageous alternative, given the limited access to novel DAAs.

#### 9.2.2. Other SSRIs

Most other SSRIs have shown antiviral mechanisms similar to fluvoxamine as they possess S1R agonism and ASM antagonism [142]. SSRIs, in general, may provide clinical benefit in preventing sCOVID-19. An initial multicenter retrospective observational study on hospitalized patients with COVID-19 found a significantly decreased risk of intubation and death in patients who were given SSRIs during hospitalization, particularly escitalopram, fluoxetine, and paroxetine [143]. A second larger-scale observational study further identified fluoxetine in addition to fluvoxamine in significantly reducing mortality with an rRR of 28.0% in patients with mCOVID-19 [144]. These results suggest that fluoxetine, like fluvoxamine, could potentially prevent COVID-19 progression in high-risk patients. Larger-scale clinical trials are required to better understand the efficacy of fluoxetine.

#### 9.2.3. Hydroxychloroquine and Azithromycin

Hydroxychloroquine was originally an antimalarial medication that was repurposed into a disease-modifying antirheumatic drug. Hydroxychloroquine has anti-inflammatory properties, attenuating production of pro-inflammatory cytokines and inflammation, which may protect against the CRS seen in sCOVID-19. Additionally, in vitro studies have shown that hydroxychloroquine could disrupt communication between SARS-CoV-2 S proteins and cell membranes, inhibiting viral entry [145]. Azithromycin is a macrolide that has been demonstrated to have in vitro antiviral activity against both Zika and rhinoviruses by upregulating anti-viral type I and III interferon responses. Azithromycin has also shown immunomodulatory activity by decreasing the hypersecretion of pro-inflammatory cytokines (e.g., IL-6, IL-1β, IFN-γ, etc.) in macrophages, monocytes, and fibroblasts [146]. The use of hydroxychloroquine and azithromycin has shown synergistic effects in inhibiting SARS-CoV-2 replication [147]. Much interest was given to hydroxychloroquine and azithromycin early in the pandemic as drugs against COVID-19 when no effective treatment and no vaccine existed. In LC patients, an early retrospective case series study observed a significantly reduced mortality rate in eight LC patients with COVID-19, treated with compassionate use of combination hydroxychloroquine–azithromycin [148]. However, hydroxychloroquine monotherapy, azithromycin monotherapy, and combination therapy for mild-to-moderate COVID-19 in high-risk patients have been evaluated in multiple phase III clinical trials but did not demonstrate significant clinical benefit in both hospitalized mCOVID-19 patients and outpatients [149,150,151]. With the advent of high-efficacy DAAs against COVID-19, use of hydroxychloroquine and azithromycin as treatment for mCOVID-19 in cancer patients outside of clinical trials should not be recommended due to the lack of evidence for clinical benefits, in addition to the potential severe adverse effects of these medications, including hydroxychloroquine-related cardiotoxicity [145]. A meta-analysis further identified increased mortality with combination hydroxychloroquine–azithromycin in COVID-19 patients [152].

### 9.3. Convalescent Plasma and Monoclonal Antibodies

#### 9.3.1. Convalescent Plasma

Convalescent plasma is an old therapy that has been historically used against multiple infectious diseases, such as MERS-CoV, H1N1, H5N1, etc. [153]. Convalescent plasma was previously studied in a phase III trial as a therapeutic for COVID-19 patients showing signs of severe pneumonia, but no significant difference was found in improving clinical outcomes or decreasing mortality between treatment and placebo groups [154]. However, a study evaluating outpatient treatment of mCOVID-19 with convalescent plasma has been recently published. The phase III trial determined that convalescent plasma significantly reduces 28-day hospitalization in mCOVID-19 patients treated with convalescent plasma versus placebo (2.9% vs. 6.3%, respectively), with an absolute RR of 3.4% (95% CI, 1.0–5.8; *p* = 0.005) [135].

#### 9.3.2. Casirivimab/Imdevimab, Bamlanivimab/Etesevimab, and Sotrovimab

Neutralizing monoclonal antibodies (nMAb) are also used to treat high-risk mCOVID-19 patients. nMAb are antibodies isolated from convalescent plasma from COVID-19 patients that can inhibit viral replication by specifically targeting and binding on epitopes on the RBD of the S protein, preventing ACE2 attachment and cell entry [155]. Cocktails of nMAbs, including casirivimab/imdevimab, bamlanivimab/etesevimab, and sotrovimab, have been evaluated in clinical trials for preventing disease progression in mCOVID-19 outpatients. The nMAbs with published results are summarized in Table 4, where they have all shown efficacy in reducing hospitalization and mortality in non-hospitalized mCOVID-19 patients, along with overall favorable safety profiles [136,137,138]. Small-scale observational reports have also shown that nMAbs are safe and have therapeutic potential in cancer patients, as outpatients with active cancer and mCOVID-19 treated with nMAbs, specifically bamlanivimab/etesevimab and casirivimab/imdevimab, demonstrated low rates of mortality and hospitalization [156,157].

Additionally, given that nMAbs are a form of passive immunization, as they involve the use of SARS-CoV-2 specific antibodies, there has been great interest in nMAbs as a form of prophylaxis in high-risk populations for SARS-CoV-2 infection. A recent phase III trial studied casirivimab/imdevimab as PoEP in healthy unvaccinated participants in close contact with an individual confirmed with SARS-CoV-2 infection. An 81.4% rRR was shown in participants exposed to SARS-CoV-2 administered with casirivimab/imdevimab. Phase 3 clinical trials for other nMAb cocktails, such as AZD7442 (tixagevimab/cilgavimab) and sotrovimab, are underway for both PrEP and PoEP in individuals at high risk of sCOVID-19 (ClinicalTrials.gov Identifiers: NCT04625725, NCT05210101, and NCT04625972). A recent paper published on the trial of AZD7442 as COVID-19 PrEP showed promising results in preventing infection for individuals at high risk for exposure or inadequate response [158].

### 9.4. Interferons

Interferons (IFN) have been well characterized as a key factor in initial immune response to viral infections, including SARS-CoV-2. Type I IFNs have an important role in restricting viral replication and spread as a key regulator of the innate and adaptive immune systems [159]. It is shown that SARS-CoV-2 can essentially escape the initial IFN response to establish viral replication, leading to asymptomatic infection and mild disease. Ineffective and delayed types I and III IFN activities, likely due to both viral escape and an impaired host immune response, are often associated with sCOVID-19 [160]. Delayed early IFN induction in COVID-19 has been shown to result in impaired T cell responses with functional exhaustion of CD4+ and CD8+ T cells and increased infiltration of pro-inflammatory cytokine-producing neutrophils and macrophages to the lungs [161]. Given the important role of dysfunctional IFN in COVID-19 progression, there has been interest in administration of exogenous IFNs as a potential treatment against COVID-19. A phase III trial on IFN β-1a showed no superior outcomes in hospitalized patients compared to placebo, with an increase in adverse events in the treatment group [162]. A strong and robust type I IFN response was observed later, in sCOVID-19, as opposed to the early delayed IFN signaling [163].

With the importance of early IFN induction during the initial stages of infection, administration of IFN may provide better benefits in the early mCOVID-19 versus during late sCOVID-19. Multiple clinical trials are underway in evaluating IFNs as a treatment for outpatients exhibiting mCOVID-19 symptoms. A phase II study found efficacy in subcutaneous pegylated type III IFN λ at accelerating viral decline in individuals with asymptomatic COVID-19 [164]. A higher probability of viral clearance by day seven was determined in pegylated type III IFN λ versus placebo (OR, 0.69; 95% CI, 0.51–0.87; *p* = 0.001). This study suggests that high-risk mCOVID-19 patients may benefit from IFNs with several trials currently underway for IFN λ as a potential treatment option for mCOVID-19 (ClinicalTrials.gov Identifiers: NCT04354259, NCT04967430, and NCT04727424). There has also been interest in using intranasal IFNs as a prophylaxis strategy, with the prophylactic potential of IFN α nasal drops observed in an open label study on hospital workers with high-risk exposure to SARS-CoV-2 [165]. Another study is currently underway evaluating prophylactic intranasal IFN γ (ClinicalTrials.gov Identifier: NCT05054114). With respect to cancer care, two ongoing studies are evaluating administration of IFNs in cancer patients against COVID-19. One group is currently studying the use of intranasal IFN α as PrEP and PoEP in cancer patients (ClinicalTrials.gov Identifier: NCT04534725) and another is evaluating intravenous IFN α and rintatolimod, a double-stranded RNA designed to mimic viral infection and stimulate immunity in cancer patients with mCOVID-19 (ClinicalTrials.gov Identifier: NCT04379518).

## 10. Discussion and Conclusions

As the COVID-19 pandemic continues into its third year, recurrent outbreaks have affected every aspect of healthcare, including the management of cancer screening, clinical investigations, and treatment.

The early consensus from cohort studies identified cancer patients as a particularly vulnerable population for sCOVID-19, with hematological malignancies and metastatic disease being identified as independent risk factors for poor COVID-19 outcomes. While studies have found conflicting results regarding LC as an independent risk factor for sCOVID-19, LC patients often harbor multiple major risk factors shared with sCOVID-19 outcomes. Additionally, immunosuppressed, and pro-inflammatory properties of the TME in LC provide the optimal environment for SARS-CoV-2 replication and spread, which can lead to diffuse cell injury, ARDS, and CRS. Several studies further characterized the following risk factors for sCOVID-19: a smoking history, older age, the male sex, multiple co-morbidities, immunosuppression, and lymphopenia. Many of these risk factors possess similar pathological mechanisms to a pro-inflammatory state and immunosuppression. Regarding the male sex as a risk factor, men with COVID-19 appear to elicit a higher inflammatory response with more pro-inflammatory cytokines and monocytes compared to women. Furthermore, CD8+ T-cell response was shown to be lower in male compared to female patients, with a negative correlation observed between older age and T-cell function observed only in men [9,10]. One study further identified lymphopenia as a significant risk factor for mortality in vaccinated cancer patients [130].

Additionally, a majority of metastatic lung cancer is under more intensive immunosuppressive sCT, leaving them at further increased risk of COVID-19 mortality [166,167]. Regarding anticancer treatments and COVID-19, the use of ICI, targeted therapies, do not lead to significantly worsened COVID-19 outcomes. sCT and RT have shown controversial findings, however. Lymphopenia secondary to sCT and RT may be the determining risk factor for poor COVID-19 outcome due to their myelosuppressive effects. Administering cytotoxic chemotherapy and RT should proceed while considering the risk of SARS-CoV-2 exposure due to their potential risk of lymphopenia. Early risk ICU assessment of cancer patients with suspected COVID-19, discussions with cancer patients and their family involving a multidisciplinary team regarding possible ICU admission, and adapted hospital protocols specifically for advanced LC patients with COVID-19 should be considered.

Clinical presentations of treatment-related toxicity and COVID-19 pneumonitis have also become a challenge in oncological care. ICI-related and RT-related pneumonitis have clinical and radiological findings that are highly similar to COVID-19 pneumonia (Table 3). Risk assessment for developing treatment-associated pneumonitis in LC patients, awareness of the differential diagnosis, and rapid COVID-19 testing strategies are crucial to accurately diagnose and treat COVID-19 pneumonia, RIP, and CIP [90,168].

During the initial waves of the pandemic, oncological care adapted to the best of its abilities to the rapidly changing circumstances of COVID-19. Oncological hubs were created as COVID-19-free centers that received cancer patients from regional hospitals overwhelmed by COVID-19 and safely continued regular cancer care, regardless of local COVID-19 prevalence. These centers were able to prevent COVID-19 outbreaks in their patient populations by following strict safety guidelines (e.g., social distancing, PPE and masks, telemedicine, SARS-CoV-2 testing prior to hospital admission, etc.) and implementing protocols in the event of suspected COVID-19. Cancer treatment regimens have also been altered to minimize hospital visits and COVID-19 exposure. For stage III NSCLC, durvalumab and RT dosing schedules were modified. An altered dose regimen, with a short fractionation regimen for RT and long interval periods for durvalumab, reduced patient exposure. In cases where certain treatments, such as surgery, were not available due to the pandemic, alternatives, such as SBRT, were proposed. While SBRT is not the standard of care for operable early-stage NSCLC compared to surgery, timely treatment of these patients with SBRT is paramount for favorable outcomes and close follow-ups should be given to patients following SBRT. Additionally, when surgical services resume, surgical resection has been proposed to patients who received SBRT if response to treatment was suboptimal following a thoracic CT scan or if patients seek further treatment [105].

Despite the efforts to adapt cancer care during the COVID-19 pandemic, high volumes of cancellations and delays in cancer screening and treatment have been observed worldwide. Routine cancer services and non-urgent surgeries were shut down for several months due to high COVID-19 infection risk in clinical settings and due to resources being redirected to combat the pandemic. Thus, healthcare providers noticed cancer-stage progression and more advanced and metastatic cancers when care services restarted, with a large backlog expected to last for more than a year. COVID-19-pandemic-related excess cancer mortality, especially from LC, is expected to significantly increase for the next few years [110,113,117,118].

Overall, patients receiving systemic cancer treatments are often immunosuppressed and may be more vulnerable to breakthrough infections [130]. A focus on therapeutic strategy for secondary prevention is required to minimize severe COVID-19 progression in cancer patients.

Multiple novel promising treatments have emerged to treat mCOVID-19 in patients at high risk for severe disease. Of all the treatments discussed, the novel DAAs paxlovid and molnupiravir, with their effectiveness in preventing sCOVID-19, favorable safety profiles, and oral route of administration, are expected to become the gold standard for treating mCOVID-19 in high-risk outpatients. Paxlovid is the preferred option due to its apparent highest efficacy. Cancer patients with a recent COVID-19 diagnosis could be prescribed with the novel DAA and be safely treated at home. Given the quicker time to recovery with the treatment of COVID-19, patients could resume life-saving anticancer treatment with minimal delays. The other treatments discussed may be good alternatives in case paxlovid is unavailable. If shown to be effective in further large-scale trials, the oral fluvoxamine, along with its accessibility and affordability, could potentially be a highly desirable alternative. Convalescent plasma may also be useful in resource-limited settings where access to antivirals is severely limited or in outbreaks of novel SARS-CoV-2 VOCs that resist most available therapies. COVID-19 prophylaxis for vulnerable cancer patients could be a potentially life-saving method for preventing sCOVID-19 outcomes and safely continue anticancer treatments, especially since they are often exposed to high-risk environments in clinics or hospitals. For all treatments discussed, there is a lack of overall data on the efficacy of these approaches against mCOVID-19, specifically in cancer patients. Most clinical trials reviewed were not able to recruit more cancer patients, where underrepresentation was a major problem. Further specific clinical trials evaluating these treatments in cancer populations are required to better investigate their efficacy.

The worldwide spread of Delta and Omicron has shown how rapidly VOCs can mutate its S proteins to increase infectivity and immune escape. These VOC mutations could decrease the efficacy of therapies that target the S protein, while further mutations might arise due to the selective pressure caused by these treatments. As currently available vaccines all generate humoral immunity targeting the S protein, vaccine breakthrough has become an increasingly difficult problem. The challenge of vaccine breakthrough is expected to increase with Omicron and future VOCs. Lymphopenia has been determined as a significant risk factor for sCOVID-19 in the studies reviewed and could have a significant role in breakthrough infections [72,79,130]. Though waning immunity was observed in cancer patients vaccinated with two doses, a third dose was able to potentiate COVID-19 immunity with high antibody titers and a strong T-cell response, even against Omicron [169,170]. Despite the increased vaccination rates in developed countries and the availability of treatments, active cancer patients, especially LC patients, remain highly vulnerable to severe COVID-19 outcomes. Exposure to SARS-CoV-2 should be minimized in this group of patients with the use of public health measures and modified cancer treatment plans during outbreak periods. Booster vaccine doses are highly recommended in immunocompromised populations, such as cancer patients, in protecting against VOCs, along with rapid treatment of early COVID-19 infection with novel antivirals to prevent severe disease outcomes.

## Figures and Tables

**Table 2 cancers-14-03598-t002:** Summary of recent anticancer treatment affecting COVID-19 outcomes.

Studies Evaluating Treatments	Treatments Evaluated	Mortality Outcomes of Cancer COVID-19 Patients
Kuderer et al. [69]	Surgery, radiation therapy, cytotoxic chemotherapy, targeted therapy, immunotherapy, endocrine therapy	No increase in mortality for all treatments evaluated.
Assaad et al. [70]	Surgery, radiation therapy, cytotoxic chemotherapy, targeted therapy, immunotherapy	No increase in mortality for all treatments evaluated.
Sharafeldin et al. [67]	Cytotoxic chemotherapy, targeted therapy, immunotherapy, endocrine therapy, hormone therapy	Increased mortality for cytotoxic chemotherapy (HR, 1.52; 95% CI, 1.1–2.1; *p* < 0.001) No significant increase in mortality for targeted, immuno-, and endocrine therapies
De Joode et al. [68]	Surgery, radiation therapy, cytotoxic chemotherapy, immunotherapy, targeted therapy	No increase in mortality for all treatments evaluated.
Ferrari et al. [71]	Radiation therapy, cytotoxic chemotherapy, targeted therapy, immunotherapy, hormone therapy	No increase in mortality for all treatments evaluated.
Chavez-MacGregor et al. [66]	Radiation therapy, cytotoxic chemotherapy, targeted therapy, immunotherapy, chemo-immunotherapy, antilymphocyte therapy	Increased mortality for radiation therapy (OR, 1.84; 95%CI, 1.28–2.31); *p* < 0.001), chemotherapy (OR, 1.84; 95% CI, 1.51–2.26, and *p* < 0.001) chemoimmunotherapy (OR, 2.31; 95% CI, 1.45–3.66; *p* < 0.001). No increase in mortality for targeted, immuno-, and endocrine therapies.
Garassino et al. [10]	Cytotoxic chemotherapy, targeted therapy, immunotherapy	No increase in mortality for all treatments evaluated.
Provencio et al. [9]	Radiation therapy, cytotoxic chemotherapy, immunotherapy.	No increase in mortality for all treatments evaluated.
Jee et al. [74]	Cytotoxic chemotherapy, targeted therapy, immunotherapy, or any combination therapy	No increase in mortality for all treatments evaluated.

**Table 3 cancers-14-03598-t003:** Clinical and radiographic findings of COVID-19 pneumonia, immune-checkpoint pneumonitis, and radiation pneumonitis.

COVID-19 Pneumonia	Immune-Checkpoint Inhibitor Pneumonitis	Radiation Therapy Induced Pneumonitis
Cough, fever, dyspnea	Cough, fever, dyspnea	Cough, fever, dyspnea
Bilateral involvement	Bilateral involvement	Unilateral, near treatment site
Ground glass opacities with reticular pattern, patchy consolidations, fibrotic and subpleural lines	Ground glass opacities, patchy peripheral consolidations, centrilobular nodules, interstitial and interseptal lobular thickening, volume loss, and traction bronchiectasis	Ground glass opacities, volume loss, linear fibrosis, consolidation, and traction bronchiectasis
High D-dimers, CRP, IL-6 and sIL-2R, and lymphopenia	High CRP, erythrocyte sedimentation, neutrophils, and lymphocytes	High CRP, erythrocyte sedimentation, ferritin and D-dimer, and lymphopenia
Can lead to ARDS	Can progress to ARDS	Can progress to ARDS

**Table 4 cancers-14-03598-t004:** Treatments with published clinical trials for mild-to-moderate COVID-19 non-hospitalized adults with high risk factors for sCOVID-19.

Treatment and Authors	Primary Endpoints	Efficacy (All Values in %)	Dosing and Route of Administration
Paxlovid-Hammond et al. [131]	28-day incidence all-cause hospitalization or mortality.	0.7 Tx vs. 6.5 placebo; rRR, 88 (*p* < 0.0001).	300 mg nirmatrelvir and 100 mg ritonavir oral BID × 5 days.
Molnupiravir-Jayk Bernal et al. [132]	29-day incidence all-cause hospitalization or mortality.	6.8 Tx vs. 9.7 placebo; rRR, 30 (*p* = 0.022).	800 mg at four 200 mg capsules oral BID × 5 days.
Remdesivir-Gottlieb et al. [133]	28-day incidence of all-cause hospitalization or mortality.	0.7 Tx vs. 5.3 placebo; hazard ratio, 0.13 (*p* = 0.008).	200 mg IV for 1 day, 100 mg IV, for next 2 days.
Fluvoxamine-Reis et al. [134]	28-day incidence of all-cause hospitalization, defined as retention in ER setting for >6 h or transfer to a tertiary hospital due to COVID-19.	11.0 Tx vs. 16 placebo; rRR, 68.0 (*p* = 0.002).	100 mg oral BID × 10 days.
Convalescent Plasma–Sullivan et al. [135]	28-day incidence of all-cause hospitalization	2.9 Tx vs. 6.3 placebo; absolute risk reduction, 3.4; 95% CI, 1.0–5.8; *p* = 0.005	250 mL transfusion over 1 h, followed by 30 min of observation
REGEN-COV- Weinreich et al. [136]	29-day incidence of all-cause hospitalization or mortality.	4.0 Tx vs. 3.2 placebo; rRR, 70.4 (*p* = 0.002).	1200 mg single dose, IV (600 mg casirivimab, 600 mg imdevimab).
Bamlanivimab/Etesevimab- Dougan et al. [137]	29-day incidence of all-cause hospitalization or mortality.	2.1 Tx vs. 7.0 placebo; rRR, 70.0 (*p* < 0.001).	5600 mg single dose, IV (2800 mg bamlanivimab/2800 mg etesivimab).
Sotrovimab Gupta et al. [138]	29-day incidence of all-cause hospitalization or mortality.	1.0 Tx vs. 7.0 placebo; rRR, 85.0 (*p* = 0.002).	500 mg single dose, IV

IV: intravenous, rRR: relative risk reduction, Tx: treatment.

## Data Availability

Data sharing not applicable. No new data were created or analyzed in this study.

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
