# Peer review of "Interactions between COVID-19 and Lung Cancer: Lessons Learned during the Pandemic"

_cancers, 2022, doi:10.3390/cancers14153598_

Round 1

Reviewer 1 Report

-line 51, he COVID-19 pandemic can be successfully modeled as a series of epidemic waves (subepidemics) and that it is possible to infer to what extent the imposition of early intervention measures can slow the spread of the disease. please discuss and cite doi:10.2196/20912

- line 56, The pandemic caused by SARS-CoV2 has stressed health care systems worldwide. The high volume of patients, combined with an increased need for intensive care and potential transmission, has forced the reorganization of hospitals and care delivery models. Standard operating procedures have been adapted both for facilities and for health care workers, including the development of well-defined and segregated patient care areas for treating those affected by COVID-19. Personal protective equipment (PPEs) availability and adequate healthcare provider training on their use should be ensured. Preventive measures are especially important in Otolaryngology-Head and Neck Surgery, as the exposure to saliva suspensions, droplets, and aerosols is increased in the upper aero-digestive tract routine examination. Moreover, the frequent invasive procedures, such as laryngoscopy, intubation, or tracheostomy placement and care, represent a high risk of contracting COVID-19. please discuss and cite doi:10.23750/abm.v92i1.11281

- line 66, The outbreak of Coronavirus disease 2019 (COVID-19) made imperative the use of protective devices as a source control tool. As there is no definite antiviral treatment and effective vaccine, the only efficient means of protecting and mitigating infectious contagion has been the use of personal protective equipment, especially by healthcare workers. However, masks affect the humidification process of inhaled air, possibly leading to a basal inflammatory state of the upper airways. please discuss and cite doi:10.7416/ai.2021.2439

- line 87, Coronavirus disease 2019 (COVID-19) is presented with asymptomatic, mild, or severe pneumonia-like symptoms. COVID-19 patients with diabetes, chronic obstructive pulmonary disease (COPD), cardiovascular diseases (CVD), hypertension, malignancies, HIV, and other comorbidities could develop a life-threatening situation. SARS-CoV-2 utilizes ACE-2 receptors found at the surface of the host cells to get inside the cell. Certain comorbidities are associated with a strong ACE-2 receptor expression and higher release of proprotein convertase that enhances the viral entry into the host cells. The comorbidities lead to the COVID-19 patient into a vicious infectious circle of life and are substantially associated with significant morbidity and mortality. The comorbid individuals must adopt the vigilant preventive measure and require scrupulous management. please cite doi:10.1016/j.jiph.2020.07.014

- Lung cancer patients have a higher mortality rate than general population. Combined hydroxychloroquine and azithromycin treatment seems like a good treatment option. It is important to try to minimize visits to hospitals (without removing their active treatments) in order to decrease nosocomial transmission. An interesting study reported 17/1878 total diagnosis in our center had lung cancer (0.9 %) versus 1878/320,000 of the total reference population (p = 0.09). 9/17 lung cancer patients with Covid-19 diagnosis died (52.3 %) versus 192/1878 Covid-19 patients in our center (p < 0.0001). Dead lung cancer patients were elderly compared to survivors: 72 versus 64.5 years old (p = 0.12). Combined treatment with hydroxychloroquine and azithromycin improves the outcome of Covid-19 in lung cancer patients, detecting only 1/6 deaths between patients under this treatment versus other treatments, with statistical significance in the univariate and multivariate logistic regression (OR 0.04, p = 0.018). please discuss and cite doi:10.1016/j.lungcan.2020.05.034

- the revision protocol needs improvements, the prisma and picot guidelines should be applied.

Author Response

Please see the attachment for the replies for the comments.

Reviewer 2 Report

The manuscript by Bian et al. focusses on an important question such as the complications of SARS-Cov-2 infection in patients with lung cancer and revises the prophylactic treatments currently available and in development.

Manuscript is very well written and covers relevant bibliography on the matter. However, before publication some minor aspects should be addressed.

-Line 87, please give the complete name of GI

-Line 247, revise the sentence, particularly the use of “of”

-Line 248, CRP is not directly produced by CD8 T cells, the sentence is misleading.

-Line 350, revise the sentence.

-Line 459, revise sentence.

-Section 4.1 includes too many data without much discussion that slows down the reading, I recommend summarizing it and to add some more discussion. As the data is in a table there is no need to be repetitive.   
